# Effect of Water Hardness on Catechin and Caffeine Content in Green Tea Infusions

**DOI:** 10.3390/molecules26123485

**Published:** 2021-06-08

**Authors:** Mica Cabrera, Faizah Taher, Alendre Llantada, Quyen Do, Tyeshia Sapp, Monika Sommerhalter

**Affiliations:** Department of Chemistry and Biochemistry, California State University East Bay, Hayward, CA 94542, USA; mcabrera15@horizon.csueastbay.edu (M.C.); ftaher@horizon.csueastbay.edu (F.T.); allantada@horizon.csueastbay.edu (A.L.); quyen2211@gmail.com (Q.D.); tyeshiasapp@gmail.com (T.S.)

**Keywords:** green tea, green tea catechins, HPLC, water hardness, epigallocatechin gallate (EGCG)

## Abstract

The health benefits of green tea are associated with its high catechin content. In scientific studies, green tea is often prepared with deionized water. However, casual consumers will simply use their local tap water, which differs in alkalinity and mineral content depending on the region. To assess the effect of water hardness on catechin and caffeine content, green tea infusions were prepared with synthetic freshwater in five different hardness levels, a sodium bicarbonate solution, a mineral salt solution, and deionized water. HPLC analysis was performed with a superficially porous pentafluorophenyl column. As water hardness increased, total catechin yield decreased. This was mostly due to the autoxidation of epigallocatechin (EGC) and epigallocatechin gallate (EGCG). Epicatechin (EC), epicatechin gallate (ECG), and caffeine showed greater chemical stability. Autoxidation was promoted by alkaline conditions and resulted in the browning of the green tea infusions. High levels of alkaline sodium bicarbonate found in hard water can render some tap waters unsuitable for green tea preparation.

## 1. Introduction

Green tea is gaining in popularity as its various health benefits, such as the prevention of cancer, cardiovascular disease, obesity, and neurodegenerative diseases, are attributed to its high content of polyphenolic compounds [1]. The four most abundant green tea polyphenols are the catechins: epigallocatechin gallate (EGCG), epigallocatechin (EGC), epicatechin gallate (ECG), and epicatechin (EC) [2]. Their corresponding stereoisomers are called gallate (GCG), gallocatechin (GC), catechin gallate (CG), and catechin (C). In vitro, green tea catechins are potent antioxidants, but their low bioavailability makes them more likely to exert their beneficial effects in vivo as signaling molecules [3]. For example, the most abundant green tea polyphenol EGCG was shown to regulate various cell signaling pathways [2]. Further health-promoting contributions of green tea polyphenols might arise after their biotransformation via gut microbiota [4].

Health-conscious consumers and researchers studying the effects of green tea consumption have a common interest in knowing and optimizing the catechin content of green tea infusions. Brewing conditions that are optimal for catechin yield, however, do not always coincide with consumers’ taste preference due to the perceived bitterness of green tea catechins [5,6]. Catechin yield depends on numerous factors, such as tea brand [7], brewing temperature and time [8], tea particle size [5], water-to-tea ratio [9], and other details of the tea preparation process [10]. Due to large variations of catechin content in green tea infusions, researchers who investigate the health benefits of green tea consumption are starting to employ supplements such as polyphenon E for a more controlled intake of green tea catechins [11].

The use of deionized or distilled water for tea preparation is most common in research studies as it enables a more reproducible tea preparation compared to the use of tap water [12,13]. Most casual tea consumers, however, will simply use their local tap water. Green tea prepared with tap or bottled mineral water was previously reported to have a lower overall catechin or EGCG content than tea prepared with distilled or deionized water [6,14,15,16]. Contrarily, Zhou and coworkers found a high EGCG content for green tea prepared with tap water [17].

Tap water can be classified into different hardness levels according to its mineral content, varying considerably by geographical origin [18]. Japan, for example, has water of low hardness (less than 90 ppm CaCO_3_ equivalents) in 92% of its region, whereas Western Europe has water of very high hardness (more than 270 ppm CaCO_3_ equivalents) in 42% of its region [19]. If water hardness influences the catechin content of green tea, consumers will have a different outcome depending on their tap water source.

Our study mimicked the tea preparation of a casual consumer who prefers the convenience of teabags, short infusion times (3 min), and lower brewing temperatures (70 °C) for a less bitter taste. We purchased synthetic freshwater in five different hardness levels for a systematic study on the influence of tap water hardness levels on green tea catechin and caffeine yield. Our controls include deionized water and solutions containing either sodium bicarbonate or the typical calcium and magnesium salts of very hard tap water. Mechanistic implications on the extraction and stability of green tea catechins and caffeine as a function of water type will be discussed.

## 2. Results

### 2.1. HPLC Quantification of Catechins and Caffeine in Green Tea Infusions

Most quantitative studies of food flavonoids, including green tea catechins, employ reverse phase C18 columns [20]. Since catechins have aromatic ring components, the use of a column with pentafluorophenyl groups (PFP) can be a useful alternative [21]. A comparison between a C18 and PFP column of traditional length (250 mm) and particle size (4 μm) is presented in the Appendix A. To reduce analysis time and solvent consumption, we switched to a shorter column (50 mm) with a small particle size (1.9 μm) and superficially porous particles. Initially, we used 0.1% formic acid as acidic modifier of the mobile phase, but GCG and CG exhibited peak tailing. A stepwise increase from 0.1% to 2% formic acid resulted in sharper and more symmetric peaks as well as shorter retention times (see Appendix A). For all subsequent HPLC experiments, the acidic modifier was kept at 2% formic acid.

Figure 1 provides sample chromatograms of the optimized method, and Table 1 summarizes HPLC performance parameters. Caffeine and all eight catechins in their epi and non-epi forms were separated from each other within a six-minute solvent gradient (see Figure 1a). The major green tea catechins, EGC, EC, EGCG, ECG, and caffeine, are readily detected in the HPLC chromatograms of green tea samples. The concentrations for the tea sample shown in Figure 1b (Lipton green tea purchased in Fall 2019) were 18.63 ± 0.79 mg per cup (235 mL) for caffeine, 32.29 ± 0.73 mg/cup for EGC, 11.17 ± 0.81 mg/cup for EC, 32.16 ± 1.69 mg/cup for EGCG, and 8.16 ± 0.40 for ECG. Peaks for GC, C, and CG were detected, but their amounts fell below the limit of quantification (LOQ). GCG was not detected.

We screened several green tea brands that we considered appealing for a casual tea consumer as these brands were affordable and readily available in caffeinated and decaffeinated versions in a typical US supermarket. HPLC data for six green tea samples (Bigelow, Lipton, and Twinnings, all purchased in caffeinated and decaffeinated versions in Fall 2017) is presented in Table 2. These teas were prepared in deionized water. The total catechin content varied from 15.12 ± 5.56 to 160.15 ± 5.47 mg per cup of green tea corresponding to a total catechin content of 8.9 ± 3.3 to 90.1 ± 3.1 mg per g of dried tea. Large variations in total catechin content for different tea brands are not uncommon [12,22,23]. The most abundant green tea catechin was EGCG, ranging from 40 to 48% of the total catechin content. All six green tea samples showed the same order of abundance for the green tea catechins: EGCG > EGC > EC > ECG > GC > C). GCG and CG were not detected in any samples, and C was not detected in three samples. The C peak detected in Bigelow and Lipton decaf green teas remained below the LOQ value of the method (1.25 mg/cup). The average caffeine content for the three caffeinated green tea samples was 25.21 ± 3.64 mg per cup. The three decaffeinated versions had an average caffeine content of 2.57 ± 0.42 mg per cup. Except for the Bigelow samples, decaffeinated versions of the same brand showed much lower total catechin content. The same trend with lower catechin contents for decaffeinated teas was also observed by Henning and coworkers [7]. Bigelow green tea is decaffeinated using a high-pressure treatment with carbon dioxide. Twinnings and Lipton green teas, however, are decaffeinated via the ethyl acetate extraction of caffeine. We therefore conclude that the process of removing caffeine via organic solvent extraction results in lower levels of green tea catechins.

### 2.2. Color and pH Changes for Green Tea Prepared with Synthetic Freshwater

For a systematic and controlled variation in water hardness, we purchased synthetic freshwater with five different hardness levels containing sodium bicarbonate, calcium sulfate, magnesium sulfate, and potassium chloride. These synthetic freshwaters represent the range of tap water hardness levels that a consumer might encounter [19,24]. Water hardness ranged from 338 ppm to 21 ppm in CaCO_3_ equivalents with conductivity values of 1025 to 41 µS/cm (Table 3). As sodium bicarbonate levels were lowered, the pH-values decreased from pH 8.3 for very hard water to pH 7.2 for very soft water. Boiling resulted in a pH increase of 0.5 to 1.0 pH units due to the release of carbon dioxide [25]. Tea brewing lowered the pH value since amino acids and other organic acids are released from the tea leaves into the water [16]. The pH values of green tea brewed with the synthetic freshwater of varying hardness ranged from pH 7.1 (very hard water) to pH 6.0 (very soft water).

Green teas prepared with very hard, hard, and moderately hard water changed color from yellow-green to dark brown (or light brown for moderately hard water) compared to samples prepared with soft and very soft water. Browning was also not observed for tea samples prepared with deionized water. UV/VIS spectra recorded for a green tea prepared with synthetic freshwater of different hardness are shown in Figure 2. The browning of the hard water teas was associated with an increase in absorbance. Wavelengths between 400 and 500 nm can be used to screen for brown compounds in tea samples [26]. We chose a wavelength of 470 nm to monitor browning. For browning to commence, the pH value and hardness level need to exceed pH 6.3 and 42 ppm CaCO_3_, respectively.

### 2.3. Dependence of Green Tea Catechin and Caffeine Content on Water Hardness

Figure 3 represents the results of the HPLC analysis conducted with green tea prepared with synthetic freshwater of different hardness. Compared to very soft water, the overall catechin content decreased from 164.6 ± 8.8 mg/cup to 67.5 ± 14.7 mg/cup for very hard water (2.4-fold). This decrease was most pronounced for EGCG and EGC with a 3.2- and 3.1-fold lower yield in very hard water preparations compared to very soft water preparations. The use of very hard water also resulted in the lowest content of ECG and EC with a 1.7- and 1.4-fold decrease compared to very soft water. In contrast, the yield of the two non-epicatechins, GC and C, was not altered by water hardness. On average, GC and C content were 10.1 ± 1.1 and 1.4 ± 0.2 mg/cup, respectively. The other two non-epicatechins, GCG and CG, were not included in this analysis as their concentrations fell below their LOQ values. Additional preparations with very hard water and other tea brands (Bigelow and Twinnings) confirmed that the overall catechin content decreases with water hardness.

Figure 3 also shows a 1.4-fold decrease in caffeine content from 27.7 ± 1.4 mg/cup (very soft water) to 19.9 ± 3.9 mg/cup (very hard water) for Lipton green tea. A decrease in caffeine content, however, was not observed for any of the other tea samples. The use of very hard water for Twinnings and Bigelow green teas resulted in caffeine content of 25.0 ± 1.5 and 22.3 ± 0.4 mg/cup, respectively. The decaffeinated versions of Lipton, Twinnings, and Bigelow had caffeine contents of 2.6 ± 0.1, 2.6 ± 0.6, and 2.9 ± 0.1 mg/cup for preparations with very hard water. These values are similar to the caffeine contents of the corresponding tea preparations with deionized water (see Table 2).

### 2.4. Which Component of Hard Water Influences Green Tea Catechin Yield?

Since very hard water contains several components and each component might influence extraction efficiency and/or chemical stability of green tea catechins in different ways, we prepared two additional solutions. One sodium bicarbonate solution to isolate the effect of alkalinity and another solution containing the remaining salts of the very hard synthetic freshwater (calcium sulfate, magnesium sulfate, and potassium chloride). Bigelow green tea (purchased in Fall 2019) was prepared with deionized water (control), the sodium bicarbonate solution, the mineral salt solution, and the very hard water of the synthetic freshwater series. To investigate the effect of oxygen, very hard water was bubbled with argon gas before and during tea preparation. The pH-values of green tea prepared with deionized water and the salt solution were 5.68 ± 0.01 and 5.51 ± 0.01, respectively. The ions Ca^2+^ and Mg^2+^ acted as Lewis acids and lowered the pH of the tea containing the salt components of very hard water. The green teas prepared with sodium bicarbonate solution, very hard water, and very hard water bubbled with argon had slightly alkaline pH values of 7.36 ± 0.03, 7.24 ± 0.08, and 7.88 ± 0.13, respectively. The presence of Mg^2+^ and Ca^2+^ ions resulted in a slightly lower pH value of the very hard water compared to the sodium bicarbonate tea preparation. The argon bubbling procedure displaced oxygen as well as carbon dioxide, which shifted the equilibrium between bicarbonate and carbonic acid (dissolved CO_2_) and resulted in the most alkaline pH value of this series.

All three tea preparations with slightly alkaline pH values showed browning in contrast to the control and mineral salt solution sample. Within the first two hours, tea prepared with sodium bicarbonate solution had the highest absorbance at 470 nm (Figure 4). As browning proceeded, the tea samples prepared with very hard water surpassed the values of the sodium bicarbonate samples. We were not able to stop browning with our argon bubbling procedure. Some residual oxygen must have remained in our tea infusions. More vigorous argon bubbling, however, would have caused a further pH increase and loss of carbonate species, since carbonic acid, the conjugated acid of carbonate, is volatile.

The HPLC analysis of the Bigelow green tea samples prepared with different water types is summarized in Figure 5. Data from the first HPLC injections (within less than one hour after sample preparation) are shown in Figure 5a. Caffeine extraction was slightly higher in more alkaline tea infusions. GC and C were detected in slightly alkaline conditions, but their contents remained below their LOQ values. GCG and CG were not detected. Among the green tea epicatechins, EGC and ECG content exhibited dependence on water type. Like caffeine, ECG had the largest yield in the sodium bicarbonate solution. The use of sodium bicarbonate solution, however, resulted in the lowest EGC content. As time progressed, EGCG displayed the same pattern as EGC, and the very hard water condition produced the lowest overall catechin content.

To assess the stability of green tea catechins, each sample was injected multiple times for HPLC analysis. As illustrated in Figure 5b, EGC and EGCG rapidly decayed in the slightly alkaline tea infusions. Approximately 3 mg EGC and 2 mg EGCG are lost per one hour in a cup of tea prepared with very hard water. In contrast, ECG and EC exhibited no significant decay. Among the non-epicatechins, only GC showed a slight positive content change (below + 0.5 mg/cup/h) in the sodium bicarbonate condition so that GC eventually surpassed its LOQ limit. A one-on-one comparison of the sodium bicarbonate and very hard water condition revealed a larger decay rate for EGC and EGCG in the presence of salt ions. This trend was not observed in a one-on-one comparison between the two acidic conditions (i.e., deionized water and mineral salt solution). The HPLC-based observations concur with UV/Vis absorption data. Initially, tea prepared with sodium bicarbonate solution showed more browning than tea prepared with very hard water. As time progressed, the presence of salts (calcium sulfate, magnesium sulfate, and potassium chloride) enhanced the brown color development in alkaline green teas. Browning hence coincides with the decline of the epicatechins EGC and EGCG.

## 3. Discussion

Most casual tea consumers will use their local tap water to prepare a tea infusion. The chemical composition and therefore, the hardness of tap water varies with season and regional source [18,27]. Since it is difficult to control and reproduce the composition of tap or mineral water, we used synthetic freshwater ranging from very soft to very hard to correlate water hardness to catechin yield. Previous studies on the effect of water quality on green tea catechin content used mineral or tap water which ranged from moderately hard to very hard. A lower total catechin content was reported for green tea prepared with tap or mineral water compared to green tea prepared with deionized or distilled water [6,14,15,16]. In agreement with our findings, EGCG and EGC yield was more affected than ECG and EC yield [14,15]. One exception is a study by Zhou and coworkers, who reported an increase in EGCG and ECG content for green tea prepared with alkaline tap and carbon adsorbed water in comparison to green tea prepared with more acidic water of low ionic strength, including deionized, distilled, reverse osmosis, and ultrapure water [17]. Since multiple factors influence the total yield and distribution of catechin species in a cup of green tea, discrepancies among studies can arise [10]. In the following, we will discuss how alkalinity and mineral salts influence catechin stability and extraction efficiency.

### 3.1. Chemical Stability of Green Tea Catechins

Three chemical processes are known to affect catechin stability: (1) autoxidation followed by polymerization, (2) epimerization, and (3) hydrolysis. Figure 6 illustrates these reactions using EGCG as an example.

The oxidation process of catechins in green tea infusions is referred to as autoxidation to emphasize that this process is not catalyzed by enzymes [26]. Green tea is unfermented tea. Enzymes such as polyphenol oxidase and peroxidase are denatured via roasting or steaming. Autoxidation of green tea catechins is more prominent in alkaline conditions [28]. EGC and EGCG are more susceptible than EC and ECG [28]. EGC and EGCG both have three hydroxyl groups attached to the B-ring of their catechin structure. This feature is called the gallyl group to distinguish it from the galloyl group attached via an ester-bond to the C-ring in ECG and EGCG. Using electron-spin resonance, Yoshioka and coworkers demonstrated that radicals form more easily on the gallyl than the galloyl moiety [29]. The radical formation is regarded as the first step of autoxidation and is followed by polymerization, which generates brown pigments [30]. The autoxidation/polymerization process clearly ties the decay of EGC and EGCG as observed via HPLC to the spectral changes observed via UV/Vis spectroscopy. This process had a major impact on our green tea infusions prepared with hard water and the sodium bicarbonate solution.

In addition to autoxidation/polymerization, green tea catechins can undergo epimerization and EGCG, or ECG could lose their galloyl moiety via hydrolysis [31]. Epimerization results in the conversion of epi-catechins into non-epi catechins (for example, EGC could turn into GC). Epimerization and autoxidation/polymerization are thought to occur in parallel, but epimerization can only be detected if it proceeds faster than autoxidation since non-epicatechins also degrade via autoxidation/polymerization [15]. In our experiments, autoxidation/polymerization dominated over epimerization. The only non-epicatechin with a barely noticeable positive content change was GC in the sodium bicarbonate infusions of Bigelow green tea. Wang and Helliwell incubated green tea infusions in hot water baths and were able to observe how epicatechins first changed into their corresponding catechins before the molecules degraded via autoxidation/polymerization [15]. High temperatures and long incubation times favor epimerization [8,15]. The loss of gallic acid via hydrolysis also requires high temperatures and ECG is less affected than EGCG [31]. As reported by Fan and coworkers, only 1.8% (or 12.9%) of EGCG molecules released gallic acid after an 8-h incubation at 60 °C (or 90 °C) [31]. We did not observe an increase in the HPLC peak of gallic acid and since our brewing temperature was only 70 °C, loss of gallic acid via hydrolysis is an unlikely decay process for EGCG or ECG in our experiments.

### 3.2. Extraction Efficiency and Complexation Reactions

The impact of water hardness on extraction efficiency is best discussed for a more stable molecule such as caffeine. More alkaline conditions might facilitate the release of compounds by rendering the cell structure of tea leaves more porous [9]. This argument would explain the small increase in caffeine for some of our very hard water tea preparations and the green tea prepared with the sodium bicarbonate solution. In contrast, Li and coworkers proposed that the formation of insoluble caffeine–catechin complexes can cause a decline in caffeine content [32]. The complex formation is promoted by hydrophobic interactions, which are enhanced in alkaline or high ionic strength conditions. It is conceivable that the formation of caffeine–catechin complexes is only noticeable for green tea preparations with very high catechin content. Lipton green tea from Fall 2017 had the highest total catechin content (Table 2) and was the only tea for which we observed a decrease in caffeine yield with increasing alkalinity (Figure 2). In general, more alkaline conditions enhanced extraction efficiency. Caffeine and the most stable catechin, ECG, had higher yields in green tea infusions prepared with sodium bicarbonate solution (pH 7.36 ± 0.03) compared to deionized water (pH 5.68 ± 0.01) as well as very hard water (pH 7.24 ± 0.08) compared to a mineral salt solution (pH 5.51 ± 0.01; Figure 5a).

Compared to the alkalinity effect of sodium bicarbonate, the effect of the salt components in synthetic hard water was more subtle. The two most stable molecules, caffeine and ECG, had higher yields for green teas prepared with deionized water compared to the mineral salt solution, as well as for the sodium bicarbonate solution compared to the very hard synthetic freshwater (Figure 5a).

Mineral salts can influence extraction efficiency in numerous ways. All plant cell walls contain the polysaccharide pectin, which readily forms complexes with Ca^2+^, Mg^2+^, and other metal ions [33]. Mossion et al. proposed that the uptake of calcium by tea leaves via complexation with pectin lowers extraction efficiency [25]. This argument was also invoked by Huang and coworkers who further argued that a high content of mineral ions could change the structure of water forming larger water clusters that are less efficient in extracting molecules from green tea leaves [14]. It was also proposed that calcium and catechins combine [34]. The increase in turbidity and sedimentation sometimes observed as “tea cream” in calcium-rich tea infusions is mostly caused by a combination of calcium ions with anions of organic acids, such as oxalate [34].

Consumers who value green tea for its high level of bioactive catechins should consider the impact of using their local tap water. Very hard water is the most unsuitable water to prepare green tea with respect to final catechin yield and the development of an ungainly brown color. The high bicarbonate content of hard water produces alkaline green tea infusions. Under these conditions, EGC and EGCG are unstable and susceptible to autoxidation followed by polymerization into brown pigments. High mineral salt levels can accelerate this degradation process and lower the extraction efficiency of bioactive molecules from tea leaves.

## 4. Materials and Methods

### 4.1. Materials

Commercial brand teabags (Lipton, Twinnings, Bigelow) in boxes of 20 were acquired at a local supermarket (Safeway, Castro Valley, CA, USA). Synthetic freshwater in five different hardness levels was from RICCA Chemical Company (Arlington, TX, USA). Green tea catechin and caffeine standards were procured from Supelco and Cerrilliant via MilliporeSigma (St. Louis, MO, USA). HPLC-grade solvents, universal pH indicator solution, and all other chemicals were obtained from Fisher Scientific Company LLC (Pittsburgh, PA, USA).

### 4.2. Tea Preparation

The chemical composition of the different water types tested in this study is summarized in Table 4. A portion of water (750 mL synthetic freshwater, deionized water, sodium bicarbonate, or mineral salt solution) was brought to boil on a hot plate in a 2 L beaker. After one minute of boiling, the beaker was removed from the hot plate, and the water was left to cool to 70 °C. A 250 mL graduated cylinder with a tolerance level of ±1.6 mL was used to measure a volume of 235 mL, close to the definition of a US customary cup (236.6 mL) [35]. Teabags were placed in Pyrex beakers of 400 mL size and immersed in 235 mL of water. After 3 min, each teabag was squeezed with a spoon, the teabag was removed, and the solution was stirred. The transfer into HPLC vials encompassed a filtration step using a syringe with a 13 mm Millex-FG filter attachment equipped with a 0.20 μm polytetrafluoroethylene (PTFE) membrane. One portion of very hard water was bubbled with argon gas for approximately 10 min and the sample preparation was conducted under a stream of argon. Three tea infusions were prepared per experimental condition.

### 4.3. HPLC Quantification of Green Tea Catechins and Caffeine

An Agilent Technologies 1260 Infinity system with autosampler was employed. An InfinityLab 120 Poroshell PFP column with dimensions of 50 × 2.1 mm and a particle size of 1.9 µm was procured from Agilent Technologies Inc. (Santa Clara, CA, USA). The detection wavelength of the diode array detector was set to 280 nm. The column temperature was controlled at 25 °C. The flow rate was set to 0.5 mL/min and the injection volume was 1 μL. Solvent A was 2% formic acid in HPLC-grade water and solvent B was acetonitrile with 2% formic acid. The following gradient was employed: 0 min (5% B, starting condition), 0–5 min (5–30% B, separation gradient), 5–6 min (30–95% B, ramp), 6–10 min (95% B, wash phase), 10–10.5 min (95–5% B, ramp), 10.5–20 min (5% B, re-equilibration).

### 4.4. UV/VIS Absorption Spectroscopy and pH Determination

UV/Vis spectra of tea samples were recorded with the cuvette option of a Nanodrop 2000c spectrophotometer from ThermoFisher Scientific (Waltham, Massachusetts, USA). To determine the pH value of tea and water samples we used a liquid universal pH indicator solution since the conductivity of several samples was too low for a traditional potentiometric measurement with a pH electrode. The pH indicator solution (50 µL) was mixed in a plastic cuvette with 3 mL of buffer, water, or tea sample. Spectra of tea samples with or without indicator were recorded from 360 to 800 nm. The spectra of the tea sample were subtracted from the spectra of the tea samples with indicator solution. The following buffers in pH increments of 0.2 units were prepared to calibrate the UV/Vis response: 0.1 M citrate—0.2 M sodium dibasic phosphate buffer in a pH range of 2.6 to 7.8, 0.1 M sodium monobasic—0.1 M sodium dibasic phosphate buffer in a pH range of 6.5 to 8.1, 0.1 M Tris-HCl buffer in a pH range of 8.0 to 9.2, and 0.1 M sodium carbonate—0.1 M sodium bicarbonate buffer in a pH range of 9.2 to 10.0.

### 4.5. Statistical Analysis

The program Minitab 17.3.1 (Minitab Inc., State College, PA, USA) was used for statistical analysis. All samples were prepared in triplicate. Significant differences of sample means were analyzed using one-way ANOVA with a significance level of α = 0.05 and the Tukey method.

## Figures and Tables

**Figure 1 molecules-26-03485-f001:**
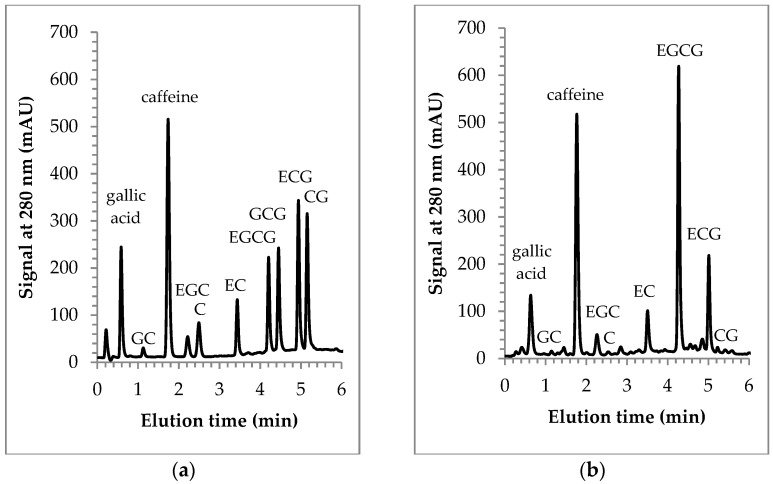
HPLC chromatograms for a standard mixture (**a**) and a green tea sample prepared with deionized water (**b**).

**Figure 2 molecules-26-03485-f002:**
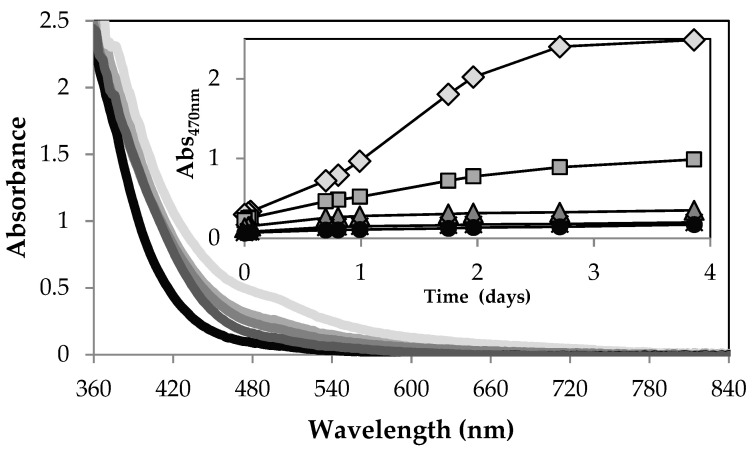
UV/Vis spectra of green tea recorded 14 h after preparation with synthetic freshwater of different hardness from very hard (light grey line) to very soft (black line). The tea brand was Lipton green tea purchased in Fall 2017. Insert: Absorbance increase at 470 nm for tea samples prepared with very soft (black circle), soft (black cross), moderately hard (grey triangle), hard (grey square), to very hard (light grey diamond) water.

**Figure 3 molecules-26-03485-f003:**
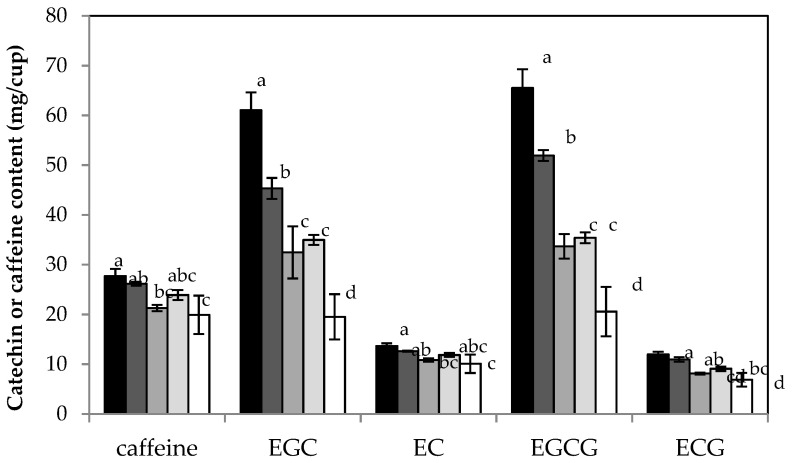
HPLC analysis of green tea (Lipton green tea purchased in Fall 2017) prepared with synthetic freshwater ranging from very soft (black columns) to very hard (white columns). The column height is the average of three tea preparations and the error bar is the standard deviation. Different letters (a–d) represent statistically different mean values for each compound compared across different water hardness levels.

**Figure 4 molecules-26-03485-f004:**
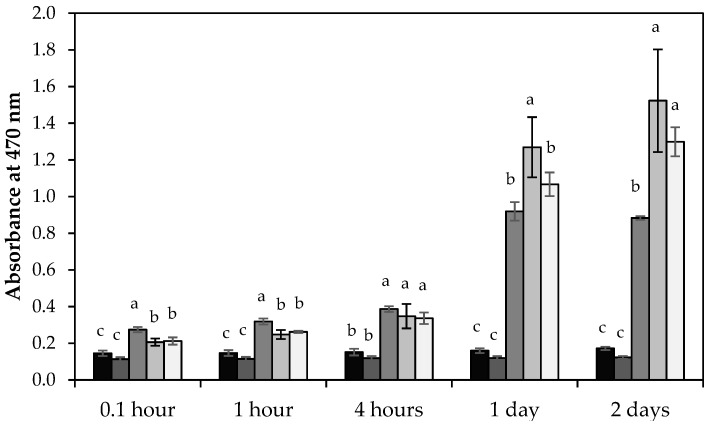
Absorbance analysis of Bigelow green tea purchased in Fall 2019 and prepared with different water types: deionized water (black), mineral salt solution (dark grey), sodium bicarbonate solution (medium grey), very hard water (light grey), and argon bubbled very hard water (white). The height of each column represents the average value and the error bar, the standard deviation of three tea preparations. Different letters (a–c) represent statistically different mean values.

**Figure 5 molecules-26-03485-f005:**
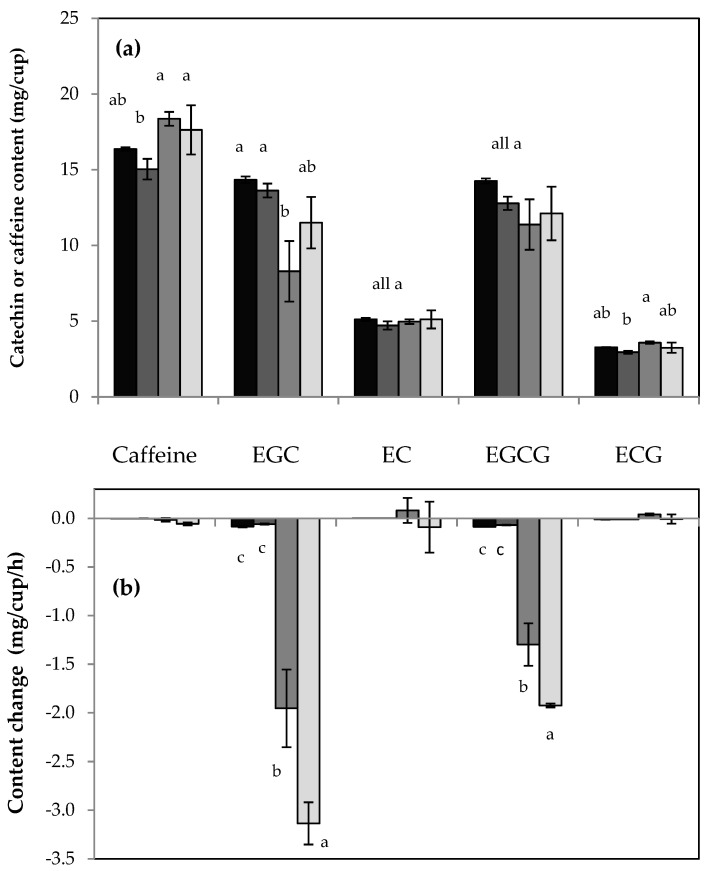
HPLC analysis of Bigelow green tea prepared with different water types: deionized water (black), mineral salt solution (dark grey), sodium bicarbonate solution (medium grey), very hard water (light grey). Data from the first HPLC run of each sample preparation is shown in (**a**) and was obtained within one hour of tea preparation. The content change obtained for a series of up to four repeated HPLC injections for 4 h is shown in (**b**). The height of each column represents the average value and the error bar, the standard deviation of three tea preparations. Different letters (a–c) represent statistically different mean values.

**Figure 6 molecules-26-03485-f006:**
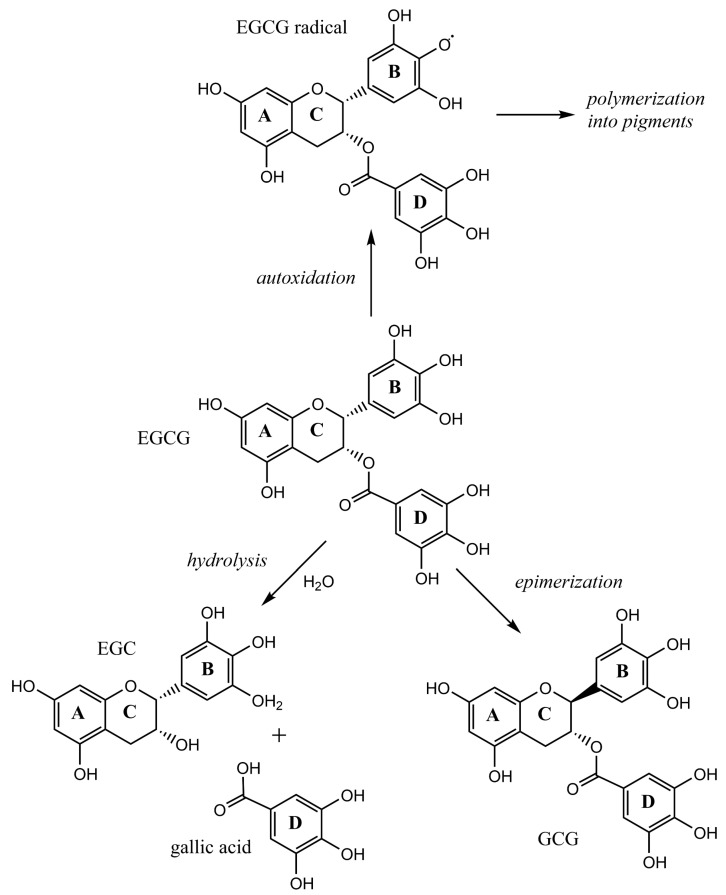
Chemical reactions that change green tea catechin content and composition.

**Table 1 molecules-26-03485-t001:** HPLC performance parameters.

Compound	Rt (min) ^1^	Resolution ^2^	Calibration Curve ^3^	R-Square ^3^	LOD ^4^ (mg/mL)	LOQ ^4^ (mg/mL)
GC	1.14 ± 0.06	-	2321 (±33) x − 1 (±1)	0.9992	0.0039	0.0130
Caffeine	1.73 ± 0.07	5.60	26,899 (±181) x − 9 (±7)	0.9997	0.0019	0.0062
EGC	2.20 ± 0.13	3.87	1620 (±14) x − 1 (±1)	0.9997	0.0024	0.0079
C	2.48 ± 0.12	2.06	8419 (±48) x − 3 (±2)	0.9999	0.0016	0.0053
EC	3.45 ± 0.09	8.24	8200 (±54) x − 3 (±2)	0.9998	0.0018	0.0061
EGCG	4.24 ± 0.06	7.90	15,779 (±191) x + 3 (±7)	0.9994	0.0034	0.0112
GCG	4.49 ± 0.06	2.51	18,004 (±144) x − 11 (±5)	0.9997	0.0022	0.0073
ECG	4.98 ± 0.06	5.03	21,064 (±122) x − 10 (±5)	0.9999	0.0016	0.0053
CG	5.19 ± 0.07	2.19	22,953 (±149) x − 12 (±6)	0.9998	0.0018	0.0059

^1^ Retention time (Rt) average and standard deviation for six determinations; ^2^ Evaluated for 0.04 mg/mL standard; ^3^ Calibration range (linear) 0.08–0.0025 mg/mL; Equation: Peak area = slope × concentration (mg/mL) + intercept; ^4^ The limit of detection (LOD = 3 *σ*_*y*0_) and limit of quantification (LOQ = 10 *σ*_*y*0_) were determined via the intercept’s standard deviation of the calibration curve, *σ*_*y*0_, with $\sigma_{y0}=\sqrt{n}\;SE_{y0}$ (*n* = 6).

**Table 2 molecules-26-03485-t002:** Green tea catechin and caffeine content in mg per cup (235 mL) for six different tea brands prepared with deionized water.

	Bigelow	Bigelow Decaf	Lipton	Lipton Decaf	Twinnings	Twinnings Decaf
GC	3.07 ± 0.06 b ^1^	below LOQ ^2^	8.58 ± 0.21 a	below LOQ	3.97 ± 0.45 b	3.60 ± 0.71 b
Caffeine	20.77 ± 1.08 b	2.24 ± 0.26 c	26.36 ± 0.45 a	3.06 ± 0.23 c	28.51 ± 1.96 a	2.42 ± 0.07 c
EGC	15.69 ± 1.13 d	17.86 ± 2.75 d	55.60 ± 1.02 a	5.86 ± 1.36 e	40.39 ± 2.81 b	25.93 ± 0.92 c
C	below LOQ	not detected	1.99 ± 0.96 a	below LOQ	not detected	not detected
EC	6.57 ± 1.49 c	6.75 ± 0.44 c	17.82 ± 1.12 a	3.28 ± 1.05 d	13.45 ± 0.62 b	10.24 ± 1.79 b
EGCG	21.38 ± 1.44 c	24.98 ± 4.24 c	64.01 ± 3.78 a	5.98 ± 5.06 d	53.92 ± 5.00 a	38.56 ± 1.11 b
ECG	4.27 ± 0.20 d	5.13 ± 0.58 d	12.14 ± 0.94 a	below LOQ	9.96 ± 1.16 b	7.54 ± 0.35 c
Total catechin	50.99 ± 1.12 d	54.71 ± 7.41 d	160.15 ± 5.47 a	15.12 ± 5.56 e	121.70 ± 9.24 b	85.88 ± 1.90 c
Teabag (g)	1.62 ± 0.03	1.65 ± 0.03	1.78 ± 0.02	1.69 ± 0.03	2.21 ± 0.04	2.11± 0.03

^1^ Each sample was prepared in triplicate. Different letters (a–e) represent statistically different mean values for individual compounds compared across different tea samples with a Tukey analysis. ^2^ The LOQ limit was 3.06 mg/cup for GC, 1.25 mg/cup for C 1.24 mg/cup for ECG.

**Table 3 molecules-26-03485-t003:** Properties of synthetic freshwater and green tea prepared with synthetic freshwater.

Synthetic Freshwater	Very Hard	Hard	Moderately Hard	Soft	Very Soft
Total dissolved solids (ppm)	830	415	207	104	25.9
Conductivity (µS/cm)	1025	560	290	155	41
Water hardness (ppm CaCO_3_ equivalents)	338	169	85	42	21
**pH values of water or tea**					
Water before boiling	8.3	8.1	7.9	7.6	7.2
Water after one minute of boiling	8.8	8.7	8.7	8.6	7.8
Green tea sample ^1^	7.1	6.8	6.5	6.3	6.0

^1^ The tea brand was Lipton green tea purchased in Fall 2017.

**Table 4 molecules-26-03485-t004:** Chemical composition of water types used in this study; all concentrations in g/L.

Water Type	NaHCO_3_	CaSO_4_ × 2 H_2_O	MgSO_4_	KCl	Source
Very soft	0.024	0.015	0.015	0.001	Synthetic freshwater (Ricca Chemical Company)
Soft	0.048	0.030	0.030	0.002	
Moderately hard	0.096	0.060	0.060	0.004	
Hard	0.192	0.120	0.120	0.008	
Very hard	0.384	0.240	0.240	0.016	
Deionized water ^1^	-	-	-	-	Prepared in the laboratory
Sodium bicarbonate solution	0.384	-	-	-	
Mineral salt solution	-	0.240	0.240	0.016	

^1^ Deionized (ultrapure, type 1) water was generated using a Millipore Direct-Q 3UV apparatus.

## Data Availability

Data is contained within the article and Appendix A.

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
