# Peer review of "Effect of Water Hardness on Catechin and Caffeine Content in Green Tea Infusions"

_molecules, 2021, doi:10.3390/molecules26123485_

Round 1
Reviewer 1 Report
Dear Authors
The objective of the manuscript “Effect of water hardness on catechin and caffeine content in green tea infusions” was to verify the effect of water hardness on catechin and caffeine content, green tea infusions were prepared with synthetic freshwater in five different hardness levels, a sodium bicarbonate solution, a mineral salt solution, and deionized water.
The article is interesting for the general public, contributing to verify how the bioactive compounds of green tea behave in waters of different hardness.
It is necessary to improve the discussion of the results, demonstrating the differential of the article. As it is presented, reading is confusing. The discussion could be divided into topics and present an illustrative scheme of the chemical reactions involved.
Page - Line 186 – Topic 2.4 not 3.4
Kind regards
Author Response
Answer to Reviewer 1:
The reviewer’s comments are colored in black. Our answers are colored in red. The line numbers are based on the final version of the revised manuscript.
Dear Authors
The objective of the manuscript “Effect of water hardness on catechin and caffeine content in green tea infusions” was to verify the effect of water hardness on catechin and caffeine content, green tea infusions were prepared with synthetic freshwater in five different hardness levels, a sodium bicarbonate solution, a mineral salt solution, and deionized water.
The article is interesting for the general public, contributing to verify how the bioactive compounds of green tea behave in waters of different hardness.
It is necessary to improve the discussion of the results, demonstrating the differential of the article. As it is presented, reading is confusing. The discussion could be divided into topics and present an illustrative scheme of the chemical reactions involved.
Page - Line 186 – Topic 2.4 not 3.4
Our answer:
We would like to thank reviewer 1 for providing several concrete ideas to improve our discussion section. We made the following four changes:
- We were asked to demonstrate the differential of the article. Previous studies on water type and green tea used bottled mineral water or tap water which is difficult to reproduce and to control due to the large regional variations of mineral and tap water. To show what is different about our work, we placed more emphasis on the advantage of using synthetic freshwater for investigating the effect of water type on green tea catechin yield at the very beginning of the discussion section (lines 257-263):
“Most casual tea consumers will use their local tap water to prepare a tea infusion. The chemical composition and therefore the hardness of tap water varies with season and regional source [18,26]. Since it is difficult to control and reproduce the composition of tap or mineral water, we used synthetic freshwater ranging from very soft to very hard to correlate water hardness to catechin yield. Previous studies on the effect of water quality on green tea catechin content used mineral or tap water which ranged from moderately hard to very hard.”
2.) We divided our revised discussion into two concrete topics with the subheadings:
“3.1 Chemical stability of green tea catechins” (see line 274)
“3.2 Extraction efficiency and complexation reactions” (see line 313)
3.) We made a new figure to illustrate the chemical reactions. Figure 6 shows the autoxidation/polymerization, hydrolysis, and epimerization reactions using the most abundant green tea catechin EGCG as an example. We inserted this new figure and the following sentence into the manuscript:
“Figure 6 illustrates these reactions using EGCG as an example.” (see line 276-277)
4.) We corrected the number of the topic 3.4 ->2.4
In addition to these changes, we specified our internal funding source (lines 420-421): “CSU East Bay” was substituted by “California State University East Bay Division of Academic Affairs” and we included the academic year of the award (2017-2018).
Reviewer 2 Report
The paper of MIca Cabrera, et al. “Effect of water hardness on catechin and caffeine content in green tea infusions” aimed to investigate water hardness effects on the catechin/caffeine content in green tea. In my opinion, it is a well-written article that deserved to be published in Molecules, but I want to make some remarks that have arisen after reading the manuscript.
- How did you choose the size of a cup? 235 mL is not a US legal cup (240 mL) or a standard 8 ounces cup (226.7 mL). Just give us the reference you used.
- The criteria of the tea brand choice are not obvious. Did you analyze three random teas?
- The physical and chemical parameters of synthetic freshwater are not discussed. How did you choose the types of water?
- The tea browning criterium is the absorbance of a tea solution at 470 nm. Why? (reference)
- You studied the influence of the various "waters" on absorbance and metabolite content in the final beverages. Only one mono-salt solution was used, the sodium bicarbonate. Why? Are you suppose that calcium sulfate, magnesium sulfate, and potassium chloride are inactive? Why?
You have to explain the "problem points" or include the references explaining your choice.
Author Response
Answer to Reviewer 2:
The reviewer’s comments are colored in black. Our answers are colored in red.
The paper of MIca Cabrera, et al. “Effect of water hardness on catechin and caffeine content in green tea infusions” aimed to investigate water hardness effects on the catechin/caffeine content in green tea. In my opinion, it is a well-written article that deserved to be published in Molecules, but I want to make some remarks that have arisen after reading the manuscript.
- How did you choose the size of a cup? 235 mL is not a US legal cup (240 mL) or a standard 8 ounces cup (226.7 mL). Just give us the reference you used.
We intended to use the US customary cup which is 8 fluid ounces or 236.6 mL. At the time that we made this decision our rationale was that a US customary cup would be a good match to the amount of water used by a casual tea consumer. The number 226.7 mL given by the reviewer is most likely a typo (or comes from the discrepancy between fluid and solid ounces, since 8 solid ounces are equivalent to 226.8 gram). In hindsight we think that using a US legal cup (240 mL) or even better a metric cup (250 mL), but not the UK imperial cup (284.13 mL), would have been better choices than aiming for the US customer cup. We used a 250 mL sized graduated cylinder with a tolerance level of +/- 1.6 mL to measure the volume of the water portions. It did not seem realistic to measure out exactly 236.6 mL, so we aimed for 235 mL to stay as close as possible to the US customer cup while reporting a volume that can be read with ease between the markings of the graduated cylinder. We added a sentence (see lines 369-371) and a reference (which lists several cup definitions) to clarify this point.
“A 250 mL graduated cylinder with a tolerance level of ± 1.6 mL was used to measure a volume of 235 mL, close to the definition of a US customary cup (236.6 mL) [35].”
New reference:
- Bucher, T.; Weltert, M.; Rollo, M.E.; Smith, S.P.; Jia, W.; Collins, C.E.; Sun, M. The international food unit: A new measurement aid that can improve portion size estimation. Int. J. Behav. Nutr. Phys. Act. 2017, 14, 1–11, doi:10.1186/s12966-017-0583-y.
- The criteria of the tea brand choice are not obvious. Did you analyze three random teas?
We added an additional sentence to clarify why we choose the three tea brands. The price (lower price range) and caffeination status might be important to a casual tea consumer. Only three brands that fulfilled our criteria were available at the local supermarket. The new sentence was inserted in line 90-92:
“We screened several green tea brands that we considered appealing for a casual tea consumer as these brands were affordable and readily available in caffeinated and de-caffeinated versions in a typical US supermarket.”
- The physical and chemical parameters of synthetic freshwater are not discussed. How did you choose the types of water?
Our manuscript already provided information on physical and chemical parameters such as pH values, dissolved solids, conductivity, and chemical composition of the synthetic freshwater used in our study (see Tables 3 and 4). However, we neglected to explicitly motivate the rationale for our choice.
Our rationale for choosing the five synthetic freshwaters was the following: Tap water is the water of choice for a casual tea consumer, but the chemical composition and physical characteristics of tap water are region-specific and will vary over time. We chose synthetic freshwater to control the composition of the waters and thereby enable a systematic and reproducible investigation. We purchased the synthetic freshwater from Ricca Chemical Company in all five different hardness levels to represent the full range of hardness levels that a consumer might encounter. The use of synthetic freshwater sets our study apart from previous studies on the effect of water quality on tea.
To clarify our rationale for using synthetic freshwater we inserted additional sentences and provided two new references:
Lines 128-129:
“These synthetic freshwaters represent the range of tap water hardness levels that a consumer might encounter [19,24].”
At the start of the discussion section (Lines 257-263):
“Most casual tea consumers will use their local tap water to prepare a tea infusion. The chemical composition and therefore the hardness of tap water varies with season and regional source [18,27]. Since it is difficult to control and reproduce the composition of tap or mineral water, we used synthetic freshwater ranging from very soft to very hard to correlate water hardness to catechin yield. Previous studies on the effect of water quality on green tea catechin content used mineral or tap water which ranged from moderately hard to very hard.”
We introduced two new references:
- US Geological Survey. Hardness of water. Available online: https://www.usgs.gov/special-topic/water-science-school/science/hardness-water? (accessed on May 24,.2021)
- Ji, Y.; Wu, J.; Wang, Y.; Elumalai, V.; Subramani, T. Seasonal Variation of Drinking Water Quality and Human Health Risk Assessment in Hancheng City of Guanzhong Plain, China. Expo. Heal. 2020, 12, 469–485, doi:10.1007/s12403-020-00357-6.
- The tea browning criterium is the absorbance of a tea solution at 470 nm. Why? (reference)
We inserted a new reference [26] and modified the text (see lines 142 - 145):
Old version:
“The browning of the hard water teas was associated with an increase in absorbance around 470-490 nm.”
New version:
“The browning of the hard water teas was associated with an increase in absorbance. Wavelengths between 400 and 500 nm can be used to screen for brown compounds in tea samples [26]. We chose a wavelength of 470 nm to monitor browning.”
New reference:
- Tan, J.; De Bruijn, W.J.C.; Van Zadelhoff, A.; Lin, Z.; Vincken, J.P. Browning of Epicatechin (EC) and Epigallocatechin (EGC) by Auto-Oxidation. J. Agric. Food Chem. 2020, 68, 13879–13887, doi:10.1021/acs.jafc.0c05716.
- You studied the influence of the various "waters" on absorbance and metabolite content in the final beverages. Only one mono-salt solution was used, the sodium bicarbonate. Why? Are you suppose that calcium sulfate, magnesium sulfate, and potassium chloride are inactive? Why?
As we studied the freshwaters composed of sodium bicarbonate, calcium sulfate, magnesium sulfate, and potassium chloride, we were most intrigued by sodium bicarbonate. This salt increased the alkalinity of the green tea solutions which turned out to be the most profound influencing factor on absorbance (see Figure 4) and green tea catechin yield/composition (Figure 5). We investigated the remaining salts as a combined mineral salt solution (a mixture of calcium sulfate, magnesium sulfate, and potassium chloride). The effect of the combined mineral salt solution was more subtle in comparison to the sodium bicarbonate solution (see lines 330-331 in our manuscript). Therefore, we did not pursue further experiments with individual mono-salt solutions.
We do not suppose that mineral salts are “inactive” (their effect is just more subtle than the effect of sodium bicarbonate). In our discussion section (see lines 334-344) we listed several means by which mineral salts can lower extraction efficiency, including complexation with pectin and change of water structure. We also cited a study by Xu and coworkers in which the authors proposed that green tea catechins and calcium ions combine (see reference [33]). To strengthen our discussion paragraph on how mineral salts can influence catechin yield in green tea infusions, we added a sentence to explain that both calcium and magnesium salts can form complexes with pectin.
“All plant cell walls contain the polysaccharide pectin which readily forms complexes with Ca2+, Mg2+, and other metal ions [33].”
New reference:
- Minzanova, S.T.; Mironov, V.F.; Vyshtakalyuk, A.B.; Tsepaeva, O. V.; Mironova, L.G.; Mindubaev, A.Z.; Nizameev, I.R.; Kholin, K. V.; Milyukov, V.A. Complexation of pectin with macro- and microelements. Antianemic activity of Na, Fe and Na, Ca, Fe complexes. Carbohydr. Polym. 2015, 134, 524–533, doi:10.1016/j.carbpol.2015.07.034.
You have to explain the "problem points" or include the references explaining your choice.
We would like to thank the reviewer for identifying specific problem points that required additional clarification and/or references. We revised the manuscript to address these shortcomings and inserted five new references.
Round 2
Reviewer 1 Report
The requests were made by the authors and the article can be accepted